# Effect of Time-Dependent Characteristics of ZnO Nanoparticles Electron Transport Layer Improved by Intense-Pulsed Light Post-Treatment on Hole-Electron Injection Balance of Quantum-Dot Light-Emitting Diodes

**DOI:** 10.3390/ma13215041

**Published:** 2020-11-09

**Authors:** Young Joon Han, Kyung-Tae Kang, Byeong-Kwon Ju, Kwan Hyun Cho

**Affiliations:** 1Manufacturing Process Platform Research and Development Department, Korea Institute of Industrial Technology (KITECH), 143 Hanggaul-ro, Sangnok-gu, Ansan-si 15588, Korea; youngjhan@kitech.re.kr (Y.J.H.); ktkang@kitech.re.kr (K.-T.K.); 2Department of Electrical and Electronics Engineering, College of Engineering, Korea University, 145 Anam-ro, Seongbuk-gu, Seoul 02841, Korea

**Keywords:** quantum-dot (QD), quantum-dot light-emitting diodes (QLEDs), intense-pulsed light (IPL), zinc-oxide nanoparticles (ZnO NPs)

## Abstract

We investigated the effect of intense-pulsed light (IPL) post-treatment on the time-dependent characteristics of ZnO nanoparticles (NPs) used as an electron transport layer (ETL) of quantum-dot light-emitting diodes (QLEDs). The time-dependent characteristics of the charge injection balance in QLEDs was observed by fabrication and analysis of single carrier devices (SCDs), and it was confirmed that the time-dependent characteristics of the ZnO NPs affect the device characteristics of QLEDs. Stabilization of the ZnO NPs film properties for improvement of the charge injection balance in QLEDs was achieved by controlling the current density characteristics via filling of the oxygen vacancies by IPL post-treatment.

## 1. Introduction

Based on the excellent electroluminescence (EL) and material properties of quantum-dots (QDs), remarkable research results for quantum-dot light-emitting diodes (QLEDs) have been reported. In addition to sharp and pure color reproducibility due to the discontinuous bandgap structure, the material stability, based on the unique properties of inorganic materials, is driving the expectation that QDs will present a new paradigm for the display industry [1,2,3,4,5]. However, QLEDs derived from organic light-emitting diodes (OLEDs) still dominate the structures in which hole transport layers (HTL) and electron transport layers (ETLs) are organic materials [6,7], and this limitation must be overcome to properly apply the inorganic material properties of QDs and improve device performance. ZnO is a representative oxide semiconductor used as a material for the ETL [8,9,10]; study of this material has been reported widely in the thin-film transistor (TFTs) research field [11,12].

In the field of QLEDs, rather than deposition of a bulk ZnO layer, research is mainly focused on the formation of an ETL in nanoparticle form by a solution process [13,14]. In this case, despite having a relatively low electron mobility compared to that of bulk ZnO, it has the advantage of being able to use low annealing that does not damage the QDs. QLEDs require a stable charge injection balance with respect to time and environmental stability. However, ZnO has a trap issue due to oxygen vacancies [15] that may negatively affect the efficiency and stability of the QLEDs. To improve the charge injection balance of QLEDs using ZnO nanoparticles (NPs) as the ETL, studies have mainly been conducted on methods of inserting buffer layers between the QD emission layer (EML) and the ZnO NPs ETL [16,17]. Based on its relatively high electron mobility compared to the hole mobility, this additional buffer layer insertion can achieve hole-electron injection balance by reducing the number of electrons injected into the QD EML. However, the insertion of an additional buffer layer is not a fundamental solution to the inherently unstable properties of ZnO NPs. In this case, trap issues in the buffer layer and the deterioration of device characteristics may be caused by the time-dependent characteristics of ZnO [18,19].

Recently, the intense-pulsed light (IPL) process has attracted much attention as an effective technique for the post-treatment of nanoparticle-based layers [20,21,22,23,24,25]. IPL, a high-intensity light emitted from a xenon lamp that produces a wide range of wavelengths from approximately 400 nm to 1200 nm in very short pulses of a few milliseconds (msec) to microseconds (μsec), enables annealing or sintering of metal [20,21,22] and oxide-semiconductor [23,24,25] nanoparticles by high-intensity light. IPL has a fast process time, which minimizes thermal damage of the device on the substrate and can achieve the post-treatment effect due to the high temperature at the layer surface.

In this study, we propose IPL post-treatment to stabilize the time-dependent characteristics of ZnO NPs. In this experiment, current density characteristic changes depending on the presence or absence of encapsulation for hole-only devices (HODs), electron-only devices (EODs), and QLEDs were measured, and the time-dependent characteristic tendencies of each QLED layer were analyzed. In addition to the stabilization of the time-dependent characteristics of devices achieved by encapsulation, the effects of IPL post-treatment of ZnO NPs on the current–voltage–luminance (IVL) characteristics of EODs and QLEDs were analyzed.

## 2. Materials and Methods

### 2.1. Materials

Poly (3,4-ethylenedioxythiophene):poly (styrene sulfonate) (PEDOT:PSS) (Clevios P VP AI 4083, Heraeus Co., Hanau, Germany) was mixed with isopropyl alcohol (IPA) at a ratio of 1:1 and sonicated at room temperature (RT) for 5 min to be used as hole injection layer (HIL) material. Next, poly [9,9-dioctylfluorene-co-N-(4-(3-methylpropyl)) diphenylamine] (TFB) (OSM Co., Goyang, Korea), used as HTL material, was dispersed in toluene (Samchun Chemical Co., Pyeongtaek, Korea) at a concentration of 8 mg/mL, and this TFB solution was stirred for 1 day at 50 °C and 400 revolutions per minute (rpm). QDs of a CdZnSeS/ZnS structure were purchased from In-visible Co. (Suwon, Korea) for use as the EML; the ligands of the QDs were formed by mixing trioctylphosphine and oleic acid. The QDs for the fabrication of QLEDs were dispersed at a concentration of 20 mg/mL in hexane. ZnO NP solution, purchased from AVANTAMA (N-11, Stäfa, Switzerland), was used as material for the ETL. The ZnO NPs having a particle size of 12 nm for the spin-coating of ETL were dispersed at a concentration of 2.5 wt% in alcohols.

### 2.2. Fabrication of HODs, EODs, and QLEDs

First, the indium-tin-oxide (ITO)-coated glass substrate was cleaned with sonication for 1 h in acetone and for 1 h in isopropyl alcohol (IPA), and ultraviolet-ozone (UV-ozone) was treated for 20 min for the uniform surface energy of the ITO-coated glass substrate. PEDOT:PSS/IPA mixed solution used as the HIL was spin-coated at 2000 rpm for 30 s on the cleaned ITO-coated glass substrate, and then annealed at 100 °C for 30 min. Next, TFB solution of a concentration of 8 mg/mL was spin-coated at 4000 rpm for 30 s and annealed at 150 °C for 30 min. TFB film was used as the HTL. After that, the QD EML was spin-coated at 5000 rpm for 30 s, and then annealed at N_2_ at 100 °C for 30 min. Next, ZnO NPs ETL spin-coating was performed at 4000 rpm for 30 s, followed by annealing at N_2_ at 60 °C for 30 min. Finally, aluminum (Al) cathodes of HODs, EODs, and QLEDs were deposited at 2 Å/s to thicknesses of 100 nm by thermal evaporation under a high vacuum pressure of 2 × 10^−7^ Torr. The HODs were fabricated with the structure of ITO/PEDOT:PSS/TFB/QD/Al by the film formation method described above, the EODs were made of the structure of ITO/QD/ZnO/Al, and the QLEDs were made of the ITO/PEDOT:PSS/TFB/QD/ZnO/Al structure. All fabricated devices were encapsulated under N_2_ atmospheric conditions.

### 2.3. Characterization and Measurements

K-alpha+ X-ray photoelectron spectroscopy (XPS) (Thermo Fisher Scientific, Waltham, MA, USA) was used to measure the binding energy properties of the ZnO NP film without and with IPL post-treatment. The QLEDs were encapsulated in an N_2_ atmospheric glove box by ejecting an XNR5570-B1 resin (Nagase ChemteX Co., Osaka, Japan) using an S-SIGMA-CM3-V5 dispenser (Musasi Engineering Inc., Tokyo, Japan) with a SHOTMASTER300ΩX tabletop robot (Musasi Engineering Inc., Tokyo, Japan). An M6100 OLED I-V-L Test System (McScience Co., Suwon, Korea) and an M6000 plus OLED Lifetime Test System (McScience Co., Suwon, Korea) were used to measure the IVL characteristics and lifetime of the QLEDs, respectively. An NX10 (Park Systems Co., Seongnam, Korea) atomic force microscope (AFM) was used to measure the surface roughness root-mean-square (RMS) values of the ZnO NP film.

## 3. Results and Discussion

Figure 1 provides a band diagram and fabrication condition of the QLEDs. The QLED band diagram in Figure 1 shows that carrier transport and injection from the lower and upper layers of the QD EML are complete, and that the HTL and ZnO NPs ETL can serve as an electron blocking layer (EBL) and hole blocking layer (HBL), respectively. Our previous work verified that the 150 °C annealing condition of TFB is suitable to prevent interlayer dissolution by organic solvents of the QD solution used to form the upper layer [26].

Figure 2 provides a schematic of the IPL post-treatment, and X-ray photoelectron spectroscopy (XPS) spectra of the ZnO NPs thin film. Figure 2a shows the oxygen vacancy (V_O_) filling effect of the IPL post-treatment. Figure 2b shows typical Zn peaks; the distances between Zn 2p_3/2_ and Zn 2p_1/2_ of the without and with IPL post-treatment were approximately 23.04 eV and 23.01 eV, respectively, values which are similar to the previously reported values [27]. These unvaried two Zn 2p peaks of Zn 2p_3/2_ and Zn 2p_1/2_ indicate that the Zn^2+^ lattice ions in the ZnO NP film are intact even after the IPL post-treatment process [28]. Oxygen-related peaks in general oxide-semiconductors are found at three binding energies [29,30]. The three peaks are the 530.2 eV peak (O-I) related to the oxygen bonded with Zn in the ZnO wurtzite structure, the 531.4 eV peak (O-II) related to the oxygen defect state (Vo) [29], and the 532.7 eV peak (O-III) related to the oxygen adsorbed on the oxide-semiconductor surface. In Figure 2c,d, we confirmed through XPS analysis that the O-II peak decreased due to the IPL post-treatment; there was a previous report indicating that this means a decrease in the V_O_ in the ZnO [29]. The increase of the O–I peak means an increase of the oxygen bonded with the Zn in the ZnO wurtzite structure, thus demonstrating that the V_O_ is filled [28,29,30].

To analyze the current density characteristics of the QLEDs, it is necessary first to understand the current behavior caused by the voltage applied to the cathode and anode electrodes [31,32]. In addition to QLEDs, stacked structures such as semiconductors, organic materials, and insulators can show current-voltage relationships in the form of a log–log plot of the current density and the applied voltage (see Appendix A, which provides graphs and concepts for the four regions).

Before applying IPL post-treatment to the device fabrication process, the current density characteristics of devices with and without encapsulation were measured to verify device stability. To analyze the time-dependent characteristics of the devices, it is important to minimize variables other than the ZnO NPs properties changed by IPL post-treatment. Using encapsulation, we have greatly mitigated the deterioration of fabricated devices during the 8-day measurement period. The deterioration of HODs due to their organic material properties has been solved, and the QLEDs’ performance does not decrease. Appendix A shows the results of improving the time-dependent characteristics of single carrier devices (SCDs) by encapsulation. In the case of QLEDs, shown in Appendix A, the stability of device performance by encapsulation was also tested.

To observe the variation of ZnO NPs film properties with IPL post-treatment, AFM analysis and current density measurement were performed. The surface roughness RMS value of ZnO NP film without IPL post-treatment, shown in Figure 3a, was 0.281 nm, indicating good uniform surface roughness properties. For the ZnO NP film with IPL post-treatment, shown in Figure 3b, the surface roughness RMS value was 0.415 nm; it seems that the reason for this increase of surface roughness RMS value is an increase of grain or nanoparticle size [20,33] of used materials due to the rapid temperature increase by thermal energy of the IPL post-treatment, or it may be due to residual solvent in the ZnO NP film. The N-11 used to form ZnO NP film in this study is an alcohol-based solution. The boiling point of alcohol is approximately 78 °C, which is approximately 18 °C higher than the 60 °C we chose for the annealing temperature of ZnO NP film. Therefore, residual solvent may remain in the ZnO NP film after the annealing process, and during the IPL post-treatment process, the residual solvent evaporates rapidly due to the rapid increase in temperature, causing solvent flow in the ZnO NP film, which may adversely affect the alignment of the ZnO NPs in the film.

Figure 3c shows the current density curve of EODs without IPL post-treatment. Over time, the current density in the ohmic current (OC) and trap space charge-limited current (T-SCLC) regions continues to decrease, which is due to the incomplete presence of V_O_ inside the ZnO NPs ETL gradually being filled with oxygen. EODs with IPL post-treatment, shown in Figure 3d, showed a great reduction in current densities in the OC and T-SCLC regions compared to the EODs without IPL post-treatment; the current density remained stable, almost unchanged until the 8-day mark. This is because sufficient thermal and photo energy to fill the V_O_ of ZnO NPs ETL with oxygen in the chamber is supplied by the IPL post-treatment. Although the surface roughness RMS value of ZnO NPs ETL slightly increased with IPL post-treatment, the current density characteristics of the EODs were successfully stabilized.

Figure 4 shows the effects of IPL post-treatment on the current density, luminance, and efficiency curve of QLEDs. As shown in Figure 4a,d, the current density of the OC and T-SCLC regions of the QLEDs decreased rapidly with IPL post-treatment; this improvement of the time-dependent characteristics of the current density is the same result as that for SCDs. Table 1 shows the summarized QLEDs’ performance values from Figure 4. Here, “stabilization time” is the time it takes for the current efficiency (CE) and external quantum efficiency (EQE) value of the measured day to reach within 96% of the measured efficiency value on the 8th day (CE of n-day / CE of 8-day < 4%). Using a comparison of CE and EQE, the stabilization time of the efficiency curve was found to have been successfully shortened by IPL post-treatment. As shown in Table 1, maximum efficiency values of the QLED with IPL post-treatment were slightly smaller than that of QLED without IPL post-treatment, because QDs were thermally degraded by the thermal and photo energy of the IPL during IPL post-treatment of the ZnO NPs ETL [34,35]. However, from immediately after fabrication to the 8th day, QLEDs showed consistently stable performance; the stabilization of the ZnO NPs ETL by IPL post-treatment helped to maintain the device characteristics.

Results of lifetime measurement of IPL post-treated QLEDs are shown in Figure 5. The device lifetime was measured in constant voltage (C-V) mode at a luminance of 1000 cd/m^2^ (driving voltage of QLEDs without and with IPL post-treatment were 3.7, and 3.9 V, respectively). Figure 5a,b shows the improvement of the hole-electron injection balance at QD EML by IPL post-treatment. Figure 5a shows that the injected electron quantity at the interface between the QD EML and the ZnO NPs ETL is larger than the injected holes’ quantity at the interface between the TFB HTL and QD EML, so that the injection balance of recombined and excited charges in the QD EML is disturbed. This imbalance of charge injection is resolved by an electron quantity decrease due to the reduced V_O_ of the ZnO NPs ETL by IPL post-treatment, as shown in Figure 5b. Figure 5c provides a graph of the lifetime of the QLEDs without IPL post-treatment. When the lifetime measurement starts, the L_T_/L_0_ value rapidly decreases for approximately 7 h, indicating lifetime deterioration due to excessive electrons accumulating at the interface between the QD EML and the ZnO NPs ETL as a result of the hole–electron injection imbalance. This decrease in L_T_/L_0_ changes to an increase in L_T_/L_0_ starting from 7 h; L_T_/L_0_ then gradually decreases after reaching its highest value. Figure 5d is the lifetime curve of QLEDs with IPL post-treatment. During the lifetime measurement, the tendency of fluctuation in the increase and decrease of L_T_/L_0_ values was solved. This phenomenon is the result of the effective control of the electron transport and injection characteristics of the ETL by oxygen-filling of the V_O_ of the ZnO NPs by IPL post-treatment, so that hole–electron injection balance was achieved. However, the lifetime (LT_50_) of QLEDs with IPL post-treatment did not increase significantly compared to QLEDs without IPL post-treatment. This may be due to the QDs having thermally degraded during the IPL post-treatment, which adversely affects the lifetime characteristics of the QLEDs. The LT_50_ of the QLEDs stabilized by IPL post-treatment was 57 h 47 min.

## 4. Conclusions

In conclusion, improvement of the stability of QLED characteristics was achieved by rapid positive-aging promoted by IPL post-treatment. After achieving device stabilization by encapsulation, the effect of IPL post-treatment on the time-dependent characteristics of ZnO NPs was analyzed. Current density values of the OC and T-SCLC regions of the EODs and QLEDs decreased rapidly from the initial state due to the positive-aging effect on the ZnO NPs ETL by IPL post-treatment; the current density curve remained steady during the measurement period of 8 days because the V_O_, which act as traps for ZnO, were sufficiently filled with oxygen from the atmosphere due to IPL post-treatment. Sufficient positive aging helped to maintain the luminance and efficiency properties of the QLEDs as well. Through the IPL post-treatment, the stabilization times for QLEDs were shortened from 1 day to initial. Finally, the effects on the time-dependent characteristics of the variation of relative luminance (L_T_/L_0_) in QLEDs were analyzed. The V_O_ of the ZnO NPs ETL was filled with oxygen by IPL post-treatment to achieve hole–electron injection balance of the QLEDs, thereby eliminating the fluctuation of L_T_/L_0_; lifetime characteristics of the QLEDs were improved.

## Figures and Tables

**Figure 1 materials-13-05041-f001:**
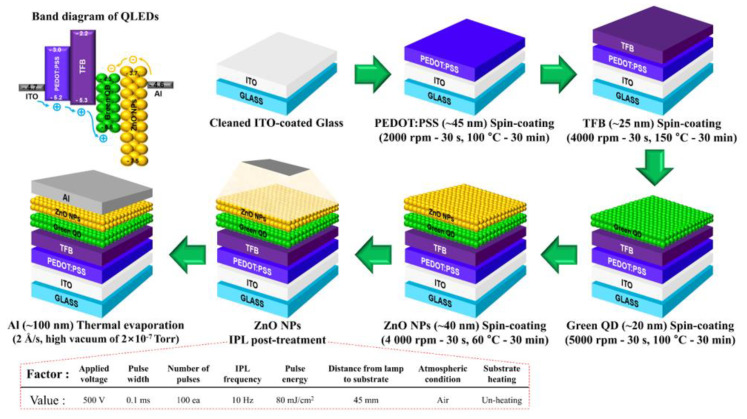
Band diagram structure and fabrication schematics for quantum-dot light-emitting diodes (QLEDs). All of the annealing process was conducted at N_2_ atmosphere.

**Figure 2 materials-13-05041-f002:**
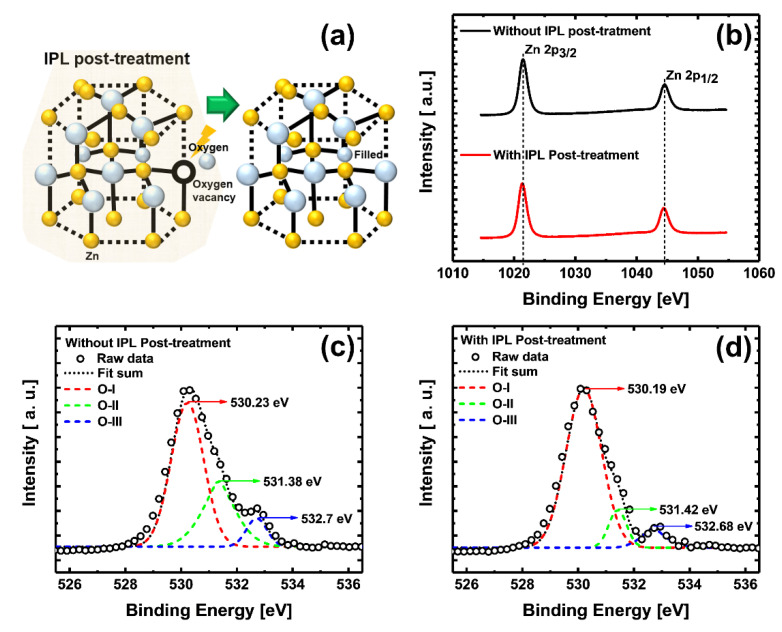
(**a**) Oxygen vacancy in the ZnO structure is filled with oxygen by intense-pulsed light (IPL) post-treatment. (**b**) Zn 2p XPS spectra of ZnO nanoparticle (NP) film (black) without and (red) with IPL post-treatment. O 1s XPS spectra of ZnO NPs film (**c**) without and (**d**) with IPL post-treatment.

**Figure 3 materials-13-05041-f003:**
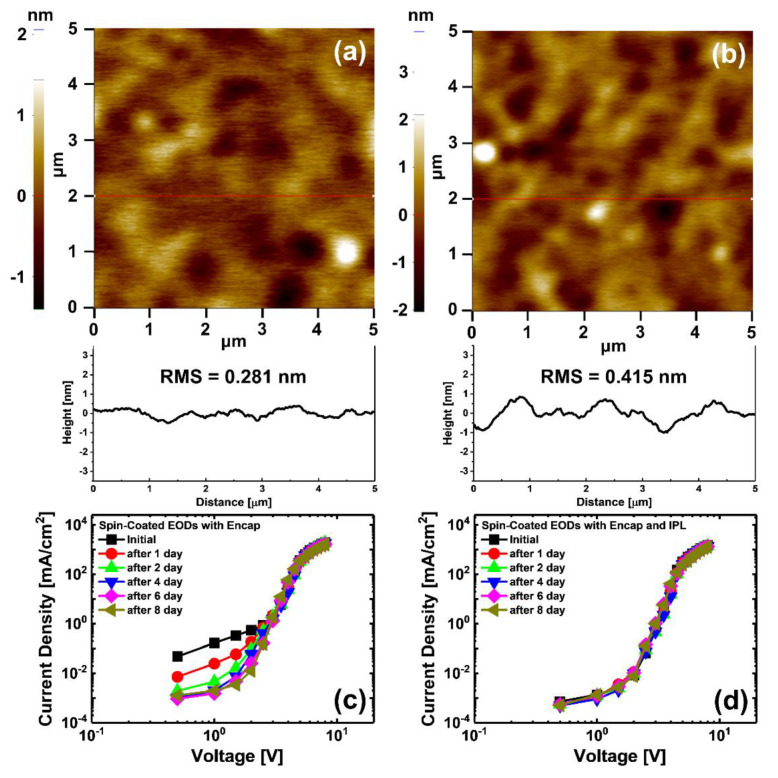
Atomic force microscope (AFM) images and line profiles of the spin-coated ZnO NPs film surface (**a**) without and (**b**) with IPL post-treatment. Current density characteristics of electron-only devices (EODs) (**c**) without and (**d**) with IPL post-treatment.

**Figure 4 materials-13-05041-f004:**
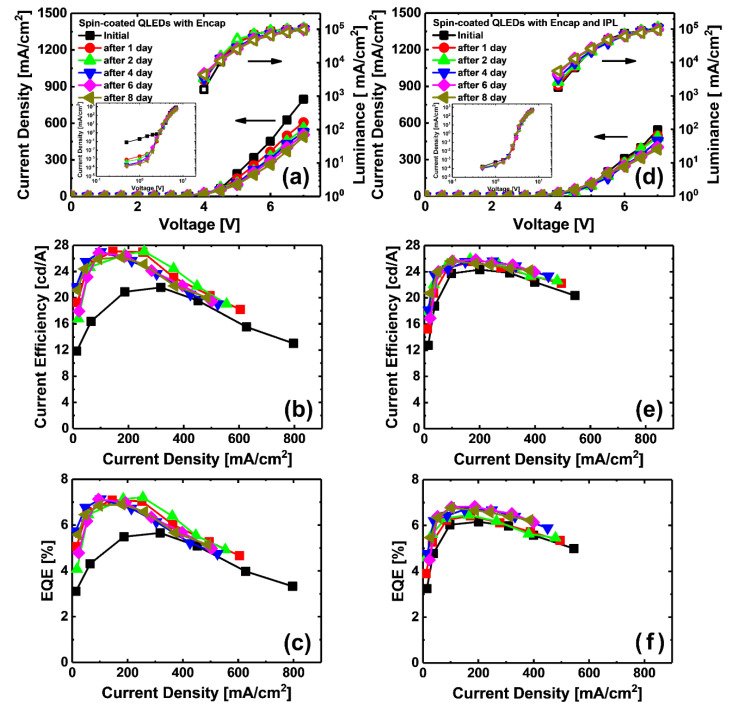
Current density and current–voltage–luminance (IVL) characteristics of the QLEDs (**a**–**c**) with and (**d**–**f**) without IPL post-treatment. (Inset of (**a**) and (**d**): log–log plot of the current density curve.)

**Figure 5 materials-13-05041-f005:**
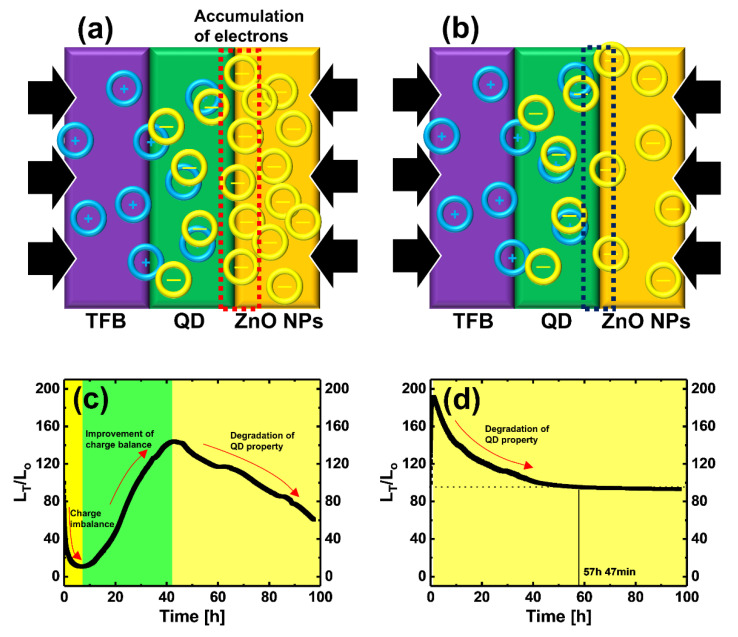
Schematics of interfaces between TFB and QD and QD and ZnO NPs of QLEDs (**a**) without and (**b**) with IPL post-treatment. Lifetime measurement of the QLEDs (**c**) without and (**d**) with IPL post-treatment.

**Table 1 materials-13-05041-t001:** The IVL characteristics and stabilization time of device time-dependency of QLEDs by IPL post-treatment.

		QLEDs without IPL	QLEDs with IPL
**Luminance (cd/m^2^)**	Initial:	103,746	110,794
	After 8 Day:	97,554	92,963
**CE (cd/A)**	Initial:	21.565	24.3235
	After 8 Day:	26.998	25.8653
**EQE (%)**	Initial:	5.657	6.159
	After 8 Day:	7.220	6.801
**Stabilization Time (Day)**		1-day	Almost Initial

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
