# Peer review of "Effect of Time-Dependent Characteristics of ZnO Nanoparticles Electron Transport Layer Improved by Intense-Pulsed Light Post-Treatment on Hole-Electron Injection Balance of Quantum-Dot Light-Emitting Diodes"

_materials, 2020, doi:10.3390/ma13215041_

Round 1

Reviewer 1 Report

The paper presents the effect of time-dependent characteristics of ZnO nanoparticles electron transport layer improved by intense-pulsed light post-treatment on hole-electron injection balance of quantum-dot light-emitting diodes. The experimental results showed useful information for future researches and the manuscript is well-organized and good description. The authors provide much understanding in the study, but more explanations and evidences for the increase in surface roughness after the IPL post-treatment in Figure 3. Overall, I recommend the publication of the paper after minor revision.

Author Response

Thank you for your kindly considered comments on our manuscript. Please find the attached MS word file, which is submitted for major revision in Materials. We hope that our response for reviewer's comments is considered favorably under your kind assistance.

Reviewer 2 Report

In this manuscript, the authors studied how intense pulsed light (IPL) post treatment can passivate ZnO nanoparticles and help stabilize the QLED I-V-L characteristics over time. This study is interesting and solid. But before this manuscript can be considered for publication in Materials, following questions and comments need to be addressed.

  1. From Figure 2, the 531.4 eV peak decreased after IPL treatment. But the peak remains obvious, which should mean that there are still a lot of oxygen defects. Will further increase the treatment duration decrease the number of defects? And how to determine whether the number of defects is small enough for the device to perform well?
  2. From Table 1, QLEDs with IPL showed a larger variation on device luminance. Could the authors explain this variation?
  3. From Table 1 and Figure 4, after stabilization, current efficiency is lower for QLEDs with IPL treatment. Does it mean IPL is not a good method? Are there any possible strategies to make IPL treatment harmless to QDs?
  4. From Figure 5, IPL treatment can help with the charge balance in a QLED device. However, are there any direct signs that IPL treatment can improve device lifetime?
  5. Some important recent progresses on QLEDs for display industry will be good to include in the background section, for example, [ Photonics 2019, 13, 192-197; J. Phys. Chem. Lett. 2019, 10, 2196-2201]

Author Response

(The authors gave the same response as above.)

Reviewer 3 Report

The authors present a study about a possible improvement of the stability of the QLED characteristics was achieved by an
Intense-Pulsed Light post-treatment.

The results are very interesting, the article is well written and well structured: the experimental procedures and the system studied are presented with many details, and the results showing an improvement in efficiency are very clear.

I have very few comments to make to the authors.
First of all, I would avoid excessive use of the English simple past, especially when describing the experiment and the system. The past perfect is much more suitable.
I would then try to use a different scale for the graphs in figure 2, especially 2b. It is not intuitive to notice the differences without and with treatment, and this being the first figure on the results seems slightly misleading.
For the rest I have no other criticisms, the article is very interesting and deserves to be published.

Author Response

(The authors gave the same response as above.)

Round 2

Reviewer 2 Report

The authors have addressed the questions and comments listed in my first review report. I recommend the revised manuscript for publication.